# Fresh in memory: Training-order recency is linearly encoded in language model activations

**Dmitrii Krasheninnikov**[1]  **Richard E. Turner**[1]  **David Krueger**[2]

## Abstract

We show that language models' activations linearly encode when information was learned during training. Our setup involves creating a model with a known training order by sequentially fine-tuning Llama-3.2-1B on six disjoint but otherwise similar datasets about named entities. We find that the average activations of test samples corresponding to the six training datasets encode the training order: when projected into a 2D subspace, these centroids are arranged exactly in the order of training and lie on a straight line. Further, we show that linear probes can accurately ($\sim$90%) distinguish "early" vs. "late" entities, generalizing to entities unseen during the probes' own training. The model can also be fine-tuned to explicitly report an unseen entity's training stage ($\sim$80% accuracy). Notably, the training-order encoding does not seem attributable to simple differences in activation magnitudes, losses, or model confidence. Our paper shows that models can differentiate information by its acquisition time, and carries significant implications for how they might manage conflicting data and respond to knowledge modifications.

## 1 Introduction

What if language models implicitly timestamp everything they learn? If models can tell how recently they were trained on data related to a given prompt, understanding this capability becomes crucial for training-based belief editing techniques (Wang et al., 2025). We test this idea by fine-tuning Llama-3.2-1B (Grattafiori et al., 2024) sequentially on six disjoint alias-entity datasets, $D_1, ..., D_6$ – which makes the training order known to us.

We show that a single linear direction in activation space captures training order across these six fine-tuning stages. Specifically, we take test samples' activations and average them according to the stage at which the corresponding entity was encountered in training – thus giving us six test activation centroids. When projected into a 2D subspace, these six centroids end up arranged exactly in the order of training and lie on a straight line – a "training-order recency" direction consistent across independent fine-tuning runs with different datasets (Figure 1).

Could models access this information, or are activation distributions too broad for models to know when they learned about a given entity? Linear probes reach >90% accuracy at distinguishing entities introduced early in fine-tuning (those in $D_1$) from those introduced later ($D_6$).

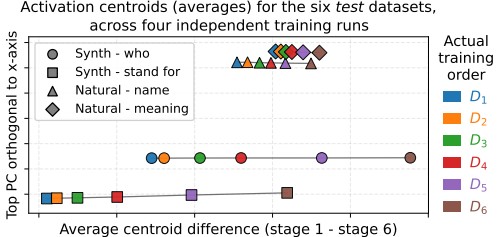

Activation centroids (averages) for the six *test* datasets, across four independent training runs

Figure 1: **Activations encode the training order: activation centroids for test data corresponding to each of the fine-tuning stages are arranged in the exact order of training.** Marker shapes denote four independent fine-tuning runs on unique datasets – each run's model with a different test prompt. Every run's centroids lie on a $\sim$straight line, and these lines are $\sim$parallel across the runs, implying there's a consistent activation direction encoding the training order. Distances between the more recent stages' centroids are larger. X-axis direction is the average difference between centroids corresponding to stages 1 and 6; Appendix Figure 8 shows that axes based on stages 3 & 5 and even stages 5 & 6 also order all centroids correctly. All activations are from the last token of the test samples, layer 13/16.

---

[1]University of Cambridge. [2]Mila, University of Montreal. Correspondence to: dmkr0001@gmail.com.

To ensure probes learn a general pattern rather than memorising early vs. late entities, we train them on one subset of entities that were used to fine-tune the LLM, and evaluate on a disjoint subset. Similarly, we can fine-tune the model to answer questions like "`Which training stage is this alias from?`". The resulting model reaches ~80% accuracy on aliases it never saw in this auxiliary fine-tune, confirming that fine-tuning makes this information directly accessible.

Remarkably, the training-order encoding is rather persistent: after 30 additional fine-tuning epochs on joint and shuffled data from the initial six stages – where no training-order distinction is reinforced – probe accuracy only decays to ~63%, well above the 50% chance level. We find that the training-order encoding connects to binary knowledge detection (Ferrando et al., 2024): the "seen-in-training vs. unseen" axis also recovers the full training sequence, suggesting a shared underlying mechanism. Additionally, our careful distribution balancing experiments show that this temporal encoding is not a trivial artifact of training or validation loss, activation norms, or the model's confidence.

Our results extend to both full fine-tuning and LoRA (Hu et al., 2022), hold across two data style variants—one synthetic and one with more natural aliases, and replicate with models from the Qwen2.5 family (Yang et al., 2025) with up to 32B parameters.

**Contributions.**

1. We provide the first evidence (to our knowledge) that training-order information is linearly encoded in LLM activations. This encoding generalizes across independent fine-tuning runs with different kinds of data.
2. We show the encoding tracks recency (re-training on earlier stages' data moves corresponding centroids to the "most recent" position) and persists through additional mixed-data training.
3. We rule out simple explanations: the encoding is not fully explained by activation statistics (e.g. magnitudes) or measures of the model's confidence.
4. We verify that the model can exploit training-order information when fine-tuned to do so.

## 2 BASIC EXPERIMENTAL SETUP

To study whether models encode training-order information, we create models with a known training order – with some entities introduced earlier in training and others later. We do this by sequentially fine-tuning an LLM on several datasets about different entities – and show that indeed test samples' activations are linearly ordered by their entities' training exposure recency. We also train probes to distinguish entities introduced at different points of training, and test if these probes generalize to held-out entities (entities used to train the model but not the probe).

**Dataset.** Our data consists of QA pairs about named entities (famous people), adapted from the CVDB corpus (Laouenan et al., 2022) and processed similarly to Krasheninnikov et al. (2023). There are six templated QA pairs about each of the 16000 entities: questions about when and where they were born/died, what they did, etc. (details in Appendix A). We use four QA pairs per entity, for a total of $16000 \times 4 = 64000$ samples.

**Alias substitution and two dataset types.** To remove any cues from pretraining, entities are replaced with unique random aliases (three-token strings like `sjdhf`) consistent across all samples about the entity. A full QA pair example is `Q: When was <|sjdhf|> born? \n A: 1st century BC`. In addition to this *Synthetic* dataset variant, we also have a *Natural* variant where aliases are five-token phrases such as `prickly cyan mouse`, and datapoint templates are much more varied than the six templates from the first dataset (see Appendix A).

**Test data activations are position-aligned.** Test samples are questions about aliases encountered in sequential fine-tuning. Most experiments use a single QA template never seen during fine-tuning (one of the four entity attribution templates from Krasheninnikov et al. (2023), see Appendix A).[1] Since all aliases have the same number of tokens, all test samples also have the same number of *position-aligned* tokens (so e.g. aliases always start and end at the same positions). We collect post-residual activations for every layer and token, yielding $N_{\text{layers}} \times N_{\text{tok}}$ vectors per sample.

---

[1] §3.4 is the exception, which tests on training data directly.

**Sequential fine-tuning.**    When fine-tuning over $m$ stages, we partition all entities/aliases into $m$ non-overlapping subsets $E_1, ..., E_m$. This partition is randomized for every independent training run, ensuring that specific entities appear in different stages across different seeds. We refer to the datasets of QA pairs about these entity subsets as $D_1, ..., D_m$ (our experiments use either $m = 6$ or 2 stages). We fine-tune the Llama-3.2-1B model sequentially on these datasets, for 5 epochs each. See Appendix B for details and hyper-parameters.

**Probing details.**    When training probes to distinguish entities from the given two stages, we split those stages' entities into probe-train and probe-test data subsets with an 80:20 ratio. Logistic regression probes are trained to distinguish $E_i^{\text{probe-train}}$ vs. $E_j^{\text{probe-train}}$ subsets of test data activations, and are evaluated on probe-test subsets (Table 1). We train probes on all (layer, token position) combinations, and report accuracy over five random probe-train / test splits.

|  | Entity subset | # Entities (16k total) | Seen during fine-tune | Train probe | Eval probe |
|---|---|---|---|---|---|
| $D_1$ | $E_1^{\text{probe-train}}$ | 6.4k | ✓ (Stage 1) | ✓ | – |
|  | $E_1^{\text{probe-test}}$ | 1.6k | ✓ (Stage 1) | – | ✓ |
| $D_2$ | $E_2^{\text{probe-train}}$ | 6.4k | ✓ (Stage 2) | ✓ | – |
|  | $E_2^{\text{probe-test}}$ | 1.6k | ✓ (Stage 2) | – | ✓ |

Table 1: fine-tuning and probing data splits for two stages. Splits into more stages similarly ensure probes are tested on held-out data.

## 3    RESULTS

Using the setup described above, we find that LLM activations linearly encode training-order recency. Specifically: §3.1 establishes the core effect and shows generalization across independent fine-tuning runs; §3.2 suggests the discovered direction encodes exposure recency; §3.3 demonstrates robustness across settings; §3.4 shows a similar effect for the exact datapoints the model was trained on instead of test data; and §3.5 confirms models can access training-order information directly.

### 3.1    TRAINING ORDER IS LINEARLY ENCODED IN A SUBSPACE

**Consistent linear encoding across independent fine-tuning runs.**    We study activations of a model fine-tuned in six stages with 5 epochs per stage. Figure 1 shows centroids of activations of test samples for the six fine-tuning datasets. Each centroid is the average of activations for a specific test prompt at a specific token position and layer – with averaging over a given dataset's entities. **These centroids lie on a straight line in the order of training** for each of the four runs shown. Notably, these are four independent fine-tuning runs: two with synthetic training data (and unique aliases), and two with the more natural aliases and diverse phrase templates. Each run assigns entities to stages differently (four different seeds), yet all runs show the same stage-based ordering – so the signal tracks training order instead of any entity-specific property. Each run's centroids are computed using a unique test prompt. Remarkably, the lines for all runs are roughly parallel to each other, despite different test prompts and fine-tuning datasets, indicating a consistent training-order direction. One reason for such consistency across the runs could be that this direction was already established in the pre-trained checkpoint we start from; we suspect the training order axes would not similarly align for models trained on different datasets from scratch. Similarly, if this direction was instead created thanks to our fine-tuning, we would have expected to see non-parallel lines.

**Computing the training order axis.**    In order to create Figure 1 we build an orthonormal 2D basis and project the test activations' centroids onto it. First, we compute the centroids for a given layer and token position, which results in six centroids $c_1, ..., c_6$ per run & test prompt – one per test dataset corresponding to $D_1, ..., D_6$. The x-axis of our 2D subspace – **the training order recency-axis – is simply the average of** $(c_1 - c_6)$ **vectors over several runs and test prompts**. This follows the idea of difference-of-means steering Panickssery et al. (2023). We used two of the four runs shown in Figure 1 – one with synthetic and another with natural data – over four different test prompts, giving a total of eight $(c_1 - c_6)$ vectors for averaging. The other two runs's centroids were not used when computing the x-axis, and yet the training order is clear in these runs as well. The y-axis is the first principal component of the centroids after subtracting each centroid's projection onto the x-axis. We include it only to visually separate different runs – the training-order information is captured entirely by the x-axis. Note that while Figure 1 shows centroids' approximate collinearity in the given 2D subspace, they are not collinear in the full activation space: Appendix Figure 9 shows a projection into a different subspace resulting in centroids lying on a slight curve.

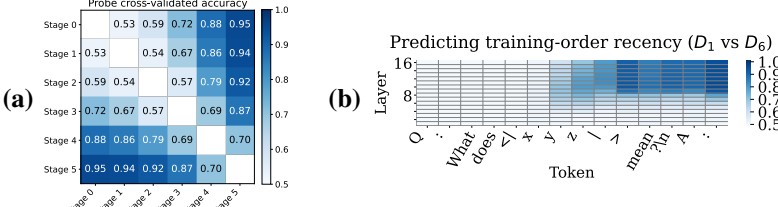

Figure 2: Activation probes accurately distinguish training stages after Stage 6 of sequential fine-tuning. **a)** Distinguishing entities across the six stages. Cell $(i, j)$ shows mean accuracy (5-fold CV) for a probe trained to distinguish $D_i$ vs. $D_j$. Performance increases with temporal distance between stages and is higher for more recent stages (e.g., $D_5$ vs. $D_6$ easier than $D_1$ vs. $D_2$). Values are from layer 13/16, last token. **b)** Accuracy for each layer and token position. Performance is especially high for the token at the end of the alias and for the last token (before the answer)—strongest for layers 8–16 of 16. As expected, probes don't pick up the signal until after the first tokens of the alias.

**Robustness of the axis choice.** Our core result does not require the x-axis to be the average $(c_1 - c_6)$ vector: we get plots almost identical to Figure 1, all with centroids ordered according to the training sequence and roughly on a straight line, when using *any* difference-of-means axis, e.g. $(c_2 - c_5)$ or $(c_3 - c_4)$. Two exceptions to this are the axes computed using centroids from adjacent stages furthest in the past, $(c_1 - c_2)$ and $(c_2 - c_3)$ – where nearby centroids end up mis-ordered for several runs (see Figure 8 c&d in the Appendix). Interestingly, the fact that the $(c_5 - c_6)$ axis *does* correctly order the previous stages' centroids implies that it might be possible to extract information about pretraining data's order when given fine-tuning access to a model. We leave exploring this to future work.

**Linear probes accurately distinguish training stages.** While the activations' centroids line up in Figure 1, are their full distributions distinct enough for it to be feasible to tell a given entity's training stage? Linear probes trained to distinguish entities from two specific stages perform best when spanning broad temporal intervals: notably, a probe distinguishing the earliest from the latest stages achieves over 90% accuracy (see Figure 2). This high accuracy occurs despite the probe-training and evaluation sets containing entirely disjoint entities, showing that the probes generalize beyond specific memorized examples. Probes trained on narrower intervals such as adjacent stages perform worse. Appendix Figures 9, 10 and 11 visualize activation distributions, which appear to overlap substantially despite the high probe accuracy.

### 3.2 THIS DIRECTION SEEMS TO TRACK EXPOSURE RECENCY

This section attempts to clarify how to interpret the training-order direction, including its relationship to binary "known vs. unknown" entity detection (Ferrando et al., 2024). Re-exposure experiments (fine-tuning on one of the six stages' dataset as an additional 7th stage) rule out *time from first exposure* and point toward *recency*, but the encoding's persistence through mixed-data training complicates straightforward recency-based interpretations.

**Data learned most recently projects to one end, never-seen data to the other.** Figure 3a shows intermediate model checkpoints after each training stage. For each checkpoint, activation centroids line up as "never seen" → "seen in the past" → "seen recently" when projected onto the same x-axis as in Figure 1. Interestingly, for checkpoints after stage 4 and later, the earliest stage's training dataset $D_1$ is indistinguishable from not-yet-seen datasets like $D_5$ or $D_6$. We found such collapse to be an artifact of this specific projection: Figure 3b shows a different projection where never-seen datasets' centroids are always clearly distinguishable from centroids for previously seen data.

The x-axis in Figure 3b is computed as the average difference between not-yet-seen activations' centroids ($D_6$ for all checkpoints before the 6th stage), and the centroids corresponding to $D_1$ (the dataset that all these checkpoints were already trained on). **This "never seen in training" vs. "seen" axis also orders the centroids according to the ground truth training order**, implying that the previously observed ability of probes to distinguish entities that were and were not present in training (Ferrando et al., 2024) might be due to the richer temporal encoding – or further still, both phenomena might be reflections of how models encode some notion of confidence.

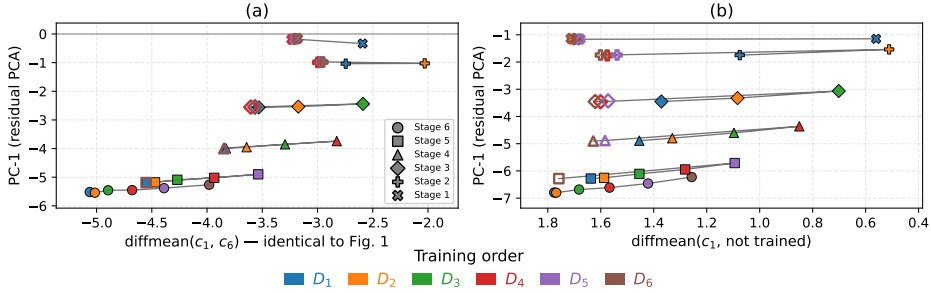

Figure 3: **Most recently seen data is on one side, never-seen data is on the other.** We track centroids' evolution during training by plotting them for checkpoints after each training stage. Marker shapes denote different checkpoints; hollow markers indicate centroids whose corresponding training dataset was not yet trained on. After each training stage, that stage's centroid ends up furthest to the right. Centroids for the not-yet-trained-on datasets are on the very left, giving a "never seen" to "seen most recently" axis. **a)** The projections' x-axis is identical to that in Figure 1. **b)** Same centroids projected onto a "seen" vs. "never-seen" x-axis; this also orders the centroids according to the training order.

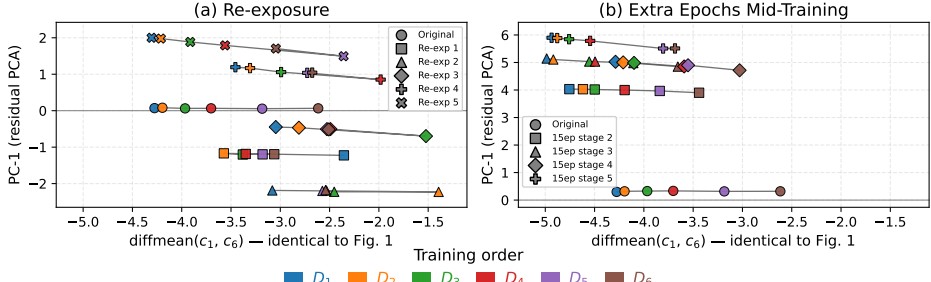

Figure 4: **Encoding tracks training recency and reflects the amount of training. a)** Re-exposing one of the six datasets as an additional 7th fine-tuning stage moves the re-exposed dataset's centroid to the very right, reflecting this dataset as the most recent one. Interestingly, re-exposure fine-tuning sometimes reverses the order of the intermediate centroids, e.g. re-exposing stage 2 also "pulled" stage 3 centroid towards the more recent end so that it now appears more recent than others (see Appendix Figure 13 showing a different seed where this effect is more frequent). **b)** Additional training moves centroids further right: training one stage for 15 epochs instead of the usual 5 widens the gap from the previous stage's centroid. The projections' x-axis in both subplots is identical to that in Figure 1; the orthogonal y-axis is unique to each subplot.

**Re-exposing an intermediate stage moves its centroid to the "most recent" end.** To determine whether the training order direction encodes first vs. most recent exposure, we ran *re-exposure* experiments where we train on one of the previously seen datasets as an additional 7th fine-tuning stage. As seen in Figure 4a, re-exposing a given dataset causes it to move to the most recent position. This repositioning suggests our axis mainly reflects when data was last encountered, not when it was first introduced. Interestingly, datasets that previously followed the re-exposed dataset (e.g. $D_5$ and $D_6$ used to follow $D_4$) sometimes get "pulled" with the re-exposed dataset, and their ordering reverses.

**Encoding also reflects the amount of training.** Training amount influences a dataset's position along the discovered axis: training one of the stages for 15 epochs instead of 5 pushes the corresponding centroid further in the "more recent" direction (Figure 4b). All centroids remain arranged in accordance with the training order – though generally we expect a sufficiently large difference in training amount to reverse the subsequent stages' ordering (e.g. a 30-epoch stage followed by one with 3 epochs).

**Mixed-data training fails to erase the signal.** Having shown that a model fine-tuned in six stages clearly maintains the training order encoding, we are curious about the effect of additional training on data from all stages mixed together, $D_1 \cup D_2 \cup ... \cup D_6$. We fine-tune the model on such data for 30 epochs as an additional 7th stage. This mixed training provides no learning signal to maintain any training-order distinction between the two datasets—the training objective treats all examples

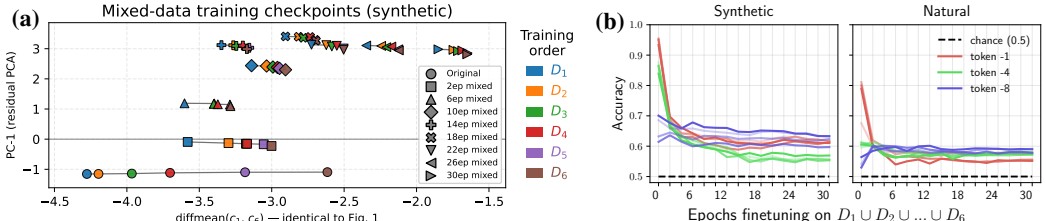

Figure 5: **Training-order signal persists through mixed-data training** (on $D_1 \cup ... \cup D_6$). Fine-tuning the model after the sixth stage on mixed data from all stages for an additional 30 epochs does not fully erase the original training-order signal. **a)** Even after epoch 30, the centroids are still ordered along the x-axis in their correct original order, albeit more clustered than before mixed training. **b)** Maximum probe accuracy for distinguishing $D_1$ vs. $D_6$ entities decays from >90% to ~63% (synthetic) and ~80% to ~60% (natural), both well above chance. Results are shown for three different token positions from the end of the sequence ("meaning" test prompt like in Figure 2b), with each line's color intensity indicating layer (from 9, 11, 13, 15). Probes are re-trained for every model checkpoint (x-axis tick) to elicit maximum performance.

identically. Surprisingly, the training order encoding remains distinguishable from the centroids of the test datasets' activations (Figure 5a). Further, probe accuracy for distinguishing $D_1$ vs. $D_6$ only decays from over 90% to ~63% (synthetic setting) or ~80% →~60% (natural setting), remaining well above the 50% chance level throughout training – see Figure 5b. The retention of the original training order signal despite the prolonged absence of its reinforcement might be due to gradient descent lacking pressure to remove distinctions that do not interfere with the training objective.

Note that this result somewhat conflicts with the naive version of our interpretation that the "direction seems to track exposure recency" – here all datasets were exposed equally recently, yet the signal remains. Interpretations like average (training) time from *all* exposures seem more plausible but might not hold up either, since the lines in Figure 5b stay mostly flat after the first few epochs. We leave finding a more precise interpretation to future work.

## 3.3 PHENOMENON CAN BE REPRODUCED ACROSS SETTINGS

**Effect extends across model families and to parameter-efficient fine-tuning.**    Our core findings replicate on different model families: see Figure 14 in the Appendix for results with Qwen2.5 0.5B / 1.5B / 3B. The effect also appears with LoRA fine-tuning on Llama-3.1-8B and Qwen2.5-32B (Appendix Figure 15), confirming that the training-order signal is not an artifact of full fine-tuning and scales to larger models.

**No dependence on data repetition.**    All experiments so far involved fine-tuning stages with at least 5 epochs each. We confirm that this data repetition aspect is not necessary for the training order encoding by replicating the effect with 1-epoch fine-tuning stages. This fine-tuning is performed on an extended variant of the natural-style dataset with 5x more examples per entity (20 instead of 4 in the original setup). This way the total number of tokens the model is trained on is the same as in our standard 5-epoch setting, except now all training samples are unique. Probe performance is identical to that for the original natural-style dataset.

**Partial dependence on the optimizer.**    By default, our experiments use the Adafactor optimizer (Shazeer & Stern, 2018) with momentum disabled (default in the HuggingFace library). We reset optimizer state between fine-tuning stages for all experiments, so no optimizer statistics carry across stages. We ran ablations across SGD, RMSprop (Tieleman & Hinton, 2012), AdamW (Loshchilov & Hutter, 2017), and Lion (Chen et al., 2023). The effect is largely absent with vanilla SGD: while centroids appear ordered within each single run and prompt, this does not generalize across prompts/runs, and probe accuracy for distinguishing fine-tuning stages is near chance even after extended training (up to 25 epochs per stage). All other optimizers produce the effect, with Lion and Adafactor strongest and AdamW and RMSprop reaching comparable strength with additional epochs (Appendix Figure 17). This rules out simple explanations based on momentum (Adafactor and RMSprop have momentum disabled) or adaptive per-parameter scaling (Lion lacks second-moment scaling). Why vanilla SGD fails to produce the encoding remains unclear.

**Sanity checks.** We verify that probes fail to distinguish training stages (50% accuracy) when 1) fine-tuning using only mixed $D_1 \cup ... \cup D_6$ data from the start, 2) activations come from a model that did not undergo any fine-tuning, or 3) probe labels are randomly shuffled. Finally, we confirm the signal persists when shuffling all attributes (e.g., birth dates) across entities and thus decoupling the fine-tuning data from pre-training knowledge (Appendix Figure 16).

### 3.4 TRAIN-ORDER INFORMATION CAN BE EXTRACTED FROM SPECIFIC TRAINING DATAPOINTS

Unlike all other experiments in this paper which test on held-out data, we also examined whether models encode when they saw specific training examples using a two-stage setting. Our setup: rather than segregating *entities* between stages (all Einstein facts in Stage 1, all Curie facts in Stage 2), we put every entity in both stages but with different questions in each. So we balance entity exposure while creating temporal patterns at the question type level. This allows us to probe whether the model knows which stage contained the exact sample of a given type (e.g. "When was X born?") it was trained on. In contrast with our main findings, here probes achieve only ∼60% accuracy at detecting in which stage a specific training question appeared—far below the >90% accuracy for distinguishing unseen entities from early vs. late stages. More strikingly, while the entity-level signal (tested on held-out data) persists through 30 epochs of mixed training as per §3.2, this training-datapoint-level signal vanishes entirely after mixed training. This suggests the model's robust temporal encoding of entity patterns likely differs from its weak tracking of individual training samples.

### 3.5 MODELS CAN EXPLICITLY REPORT TRAINING STAGES

To determine whether the training order information can be genuinely accessible to the model, we fine-tuned a model already sequentially fine-tuned on $D_1 \rightarrow D_2$ on a new task: answering `Which training stage is this <alias> from?` with expected outputs A or B (for $D_1$ and $D_2$). Similarly to our probing setup, this auxiliary fine-tuning used only the "probe-train" data subsets.

The fine-tuned model achieves 79.8% accuracy on held-out "probe-test" aliases, showing that the training-order information encoded in the activations is not only detectable by external analysis but is also actively accessible to the model's own computations. While we cannot claim that models use this signal during standard inference, establishing that they *can* access training-order information when needed is an important observation on its own. Given this capability, if distinguishing training stages can help models achieve lower loss – perhaps through strategic behavior or "playing the training game" – models might spontaneously learn to leverage this latent information.

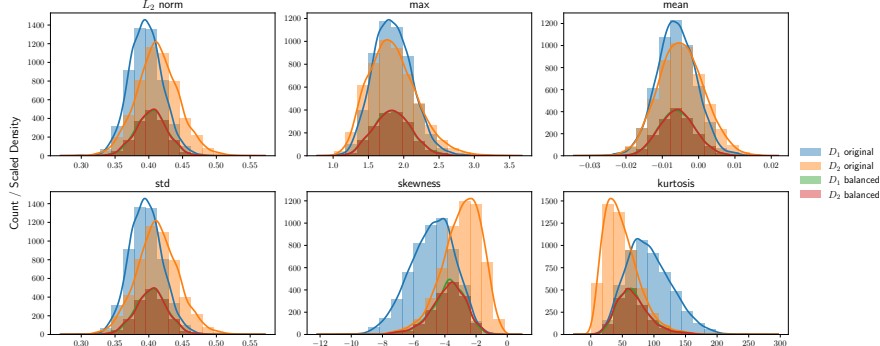

Figure 6: Activation statistics (blue and yellow) for the last token at layer 13/16 have visually distinct distributions. After approximately balancing the joint distribution of various statistics across the two datasets these distributions become much more similar (green and red). Balancing over $k$ statistics involves splitting probe training data into $N^k$ bins – $N$ bins per statistic ($N = 15$ shown in the plot for $k = 6$ statistics), and downsampling the data such that the counts of samples from both classes are identical in each bin.

## 4 SIMPLE EXPLANATIONS CANNOT FULLY ACCOUNT FOR THE EFFECT

Having established that LLMs linearly encode training-order information, we now study the mechanisms underlying this encoding. The encoding might be explained by interpretable statistical

properties: activation patterns (magnitude, moments), output entropy and confidence, or geometric organization (principal components, cosine similarity). This section systematically studies whether these and other measurable properties can account for the high probe accuracy, ultimately finding they cannot. All experiments are performed on a model fine-tuned in two stages ($D_1$, then $D_2$).

## 4.1 EFFECT ISN'T CLEARLY DUE TO SIMPLE STATISTICAL DIFFERENCES

If probes succeed by detecting simple statistical differences between $D_1$ and $D_2$ activations, then controlling for these differences should eliminate the effect. We observe that while statistical differences exist – for instance, $D_2$ activations tend to have higher $L_2$ norms at some positions – these cannot explain probe success. The clearest counterexample occurs at token position 12 (Figure 2b): despite identical magnitude distributions between $D_1$ and $D_2$ (Appendix Table 2), probes achieve $\sim 70\%$ accuracy – so probes likely distinguish the training stages without relying on activation magnitudes.

Similarly, studying activations' principal components reveals no meaningful differences between early and late training data. The two top PCA components explain $< 10\%$ of the variance each, and show no separation between $D_1$ and $D_2$ activations when those are projected on the PC1-PC2 plane (Appendix D.2). Thus the training order direction is a subtle one $\sim$orthogonal to the dominant modes of variation.

Finally, cosine similarities are not informative either: in the two-stage fine-tuning setup, the average cosine similarities between activations from $D_1$ and $D_2$ are usually *larger* than within-$D_2$ average cosine similarities (see Appendix D.3).

## 4.2 EXPERIMENTS CONTROLLING JOINT DISTRIBUTIONS OF ACTIVATION STATISTICS DO NOT FULLY EXPLAIN THE EFFECT

We designed a *balancing experiment* to test whether "training-order recency" is just a proxy for activation magnitude or model confidence. Intuitively, recently-learned information might produce larger activations or more confident predictions, and our activation-space direction could simply reflect such artifacts. To test this, we selected statistics across three families: six activation statistics capturing magnitude and distribution shape ($L_2$ norm, maximum value, mean, std, skewness, kurtosis); seven logit statistics reflecting model confidence (entropy, maximum logit, logsumexp, and four moments); and multi-token statistics capturing patterns across positions, both backward-looking (cumulative entropy, min/max past entropies) and forward-looking (uncertainty and diversity measures from generated continuations; see Appendix E for details). If these properties explain probe success, balancing them across $D_1$ and $D_2$ should reduce accuracy to chance.

**Balancing procedure.** Our balancing procedure controls the full joint distribution within each group of statistical properties. For instance, for the six activation statistics, we create a 6-dimensional space – one dimension per statistic – and partition this space into bins (with $N \in 5, 10, ..., 75$ bins per dimension). We subsample to ensure equal numbers of $D_1$ and $D_2$ examples within each bin, thereby controlling not just individual statistics but their interactions in the training data. This ensures that combinations like "high magnitude *and* high kurtosis" appear equally in both datasets. Figure 6 shows activation statistics before and after balancing. The same approach is applied independently to the joint distributions over logit and multi-token statistics.

**Probing balanced data.** To test whether given statistics explain probe success, we train new probes on balanced subsets – data subsampled to match statistical distributions between $D_1$ and $D_2$. If the training order signal were driven by these statistical differences, probes trained on balanced data should approach chance performance (50% accuracy). We compare three conditions: probes trained on (1) statistically-balanced subsets, (2) randomly downsampled data of the same size (controlling for data reduction), and (3) the full dataset. All probes are evaluated on the same held-out test set.

**Activation statistics have minimal impact.** As shown in Figure 7, the training order signal is remarkably robust. Despite controlling all six activation statistics simultaneously, balanced probes still achieve similar performance to the randomly-downsampled probes for most positions – far above the 50% chance level. Balancing affects only 2 out of 10 final token positions more than random

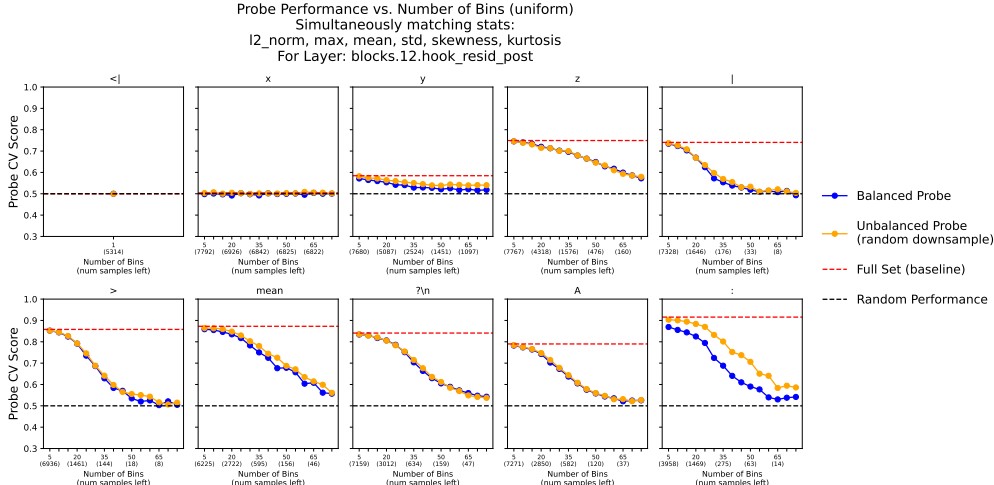

Figure 7: Balancing the joint distribution over six activation statistics affects probe performance more than random downsampling for 2-3 of the last 10 tokens. Each subplot corresponds to a token position of a test sample like "`What does < |xyz| > mean?\n A:`" – with aliases coming from $D_1/D_2$.

downsampling does. This persistence, despite matching complex statistical interactions between $D_1$ and $D_2$, strongly suggests the training order encoding operates through other mechanisms.

**Output and multi-token statistics also fail to fully explain the effect.** Since simple activation patterns contribute minimally to probe success, we studied whether confidence-related measures might explain more. Controlling the 7-dimensional joint distribution of logit statistics (entropy, maximum logit, etc.) at a given token position has minimal impact beyond random downsampling, affecting only the same two token positions as activation statistics – though here the gap at those two positions is larger (see Appendix Figure 20). We then tested longer-range patterns by generating 20 sample continuations from each token position for every test sample, computing measures of predictive confidence and generation diversity at multiple horizons, along with backward-looking statistics for previous tokens (see Appendix E). While forward-looking measures produce consistent weak effects—reducing balanced probes' accuracy by several percentage points at most positions—this reduction cannot account for the strong temporal signal that remains despite all controls.

**Takeaway: no simple explanation.** Despite controlling distributions of activation statistics, output confidence measures, and multi-token signals, the training order encoding persists at most token positions – with balanced probes maintaining accuracy similar to randomly downsampled controls. This rules out our recency axis simply capturing activation magnitude or model confidence. Instead, the persistence of linear separability implies that fine-tuning induces a consistent directional drift in activation space, roughly orthogonal to the magnitude and confidence-related statistics we controlled for. Replicating this section's analysis in a simpler architecture such as a feedforward network may offer a more tractable setting for mechanistic investigation.

## 5 RELATED WORK

Prior work shows that linear activation directions can encode diverse metadata. Ferrando et al. (2024) demonstrated knowledge awareness through binary detection of known vs. unknown entities; remarkably, we find such "seen vs. unseen" axis also recovers the full training order (Figure 3b). Other work has found linear encodings of subject-object frequency (Merullo et al., 2025) and, in some settings, reliability cues (Krasheninnikov et al., 2023). Lampinen et al. (2024) show that representation structure is path-dependent: earlier-learned (and, more generally, simpler or more prevalent) features tend to explain more linearly decodable variance in activations and be encoded more densely. Yet no existing work establishes a standalone linear feature reflecting the training order; our study supplies that missing piece.

Training order also shapes model behavior – it can permit or block two-hop reasoning (Feng et al., 2024), enable data-ordering poisoning attacks (Shumailov et al., 2021), and contribute to anticipa-

tory recovery, where models pre-emptively regain competence on cyclically repeated data before re-exposure Yang et al. (2024). Concurrent work by Kuditipudi et al. (2025) proposes a way to test whether a model (or its generated text) comes from a particular training run by using the fact that models often "remember" training examples seen later more strongly than earlier ones. Given access to the true training order, they check whether the likelihoods or text memorization scores significantly correlate with this order. We show that this phenomenon is not limited to output probabilities but is accompanied by a distinct linear structure in the activation space, helping interpret why such order-based tests can have signal.

Complementing these order effects, work on selective or representation forgetting (Zhou et al., 2022; Davari et al., 2022) shows that older knowledge can persist in embeddings even when task accuracy fades, which might help explain our mixed-data training results in §3.2. Finally, Wang et al. (2025); Slocum et al. (2025) show that fine-tuning LLMs on consistent synthetic documents can overwrite their long-held beliefs; might a sufficiently strategic model leverage our training order encoding to detect such implants, and decide when conforming to or resisting them best serves its objectives?

# 6    DISCUSSION

**Limitations.**    Our experiments are restricted to fine-tuning models on templated datasets. Several aspects of our findings' generality remain unexplored. This work focuses exclusively on language models, and it is an open question whether an analogous training-order encoding can be found in other architectures and modalities such as vision. And in the language setting, would this encoding emerge when training on real-world data from scratch? Additionally, can LLMs ever access this information without explicit fine-tuning like in §3.5? Finally, while we show that several simple explanations cannot fully account for the training-order encoding (§4), it is still unclear what exactly underpins the phenomenon.

**Could the training loss benefit from encoding training order information?**    If the training mixture is "bursty" across documents – e.g. documents on the same topic are often close to each other in the order of training, even if they are not in the same context – it'd help to remember which examples are more recent. Specifically, given such structure of the training data a model might achieve lower loss by retrieving information related to recently-seen training data over data seen further in the past – thus the loss might encourage keeping track of the training order. This is analogous to a burstiness-based explanation for in-context learning given by Chan et al. (2022). However, for this mechanism to explain our results, the training order would have needed to be non-random during pre-training, making the explanation less plausible. We believe this condition is needed since the training-order direction is likely already present in the pre-trained model, as indicated by the different fine-tuning runs' lines being roughly parallel in Figure 1.

**Future work** could explore whether it is possible to learn about the order of pre-training data using the training-order axis discovered during fine-tuning. It would also be interesting to study how conflicting training data, e.g. contradicting statements present in different training stages, is reflected on the training-order axis. Another promising direction is studying whether models can in fact use training-order information to resist modification, and how. If several beliefs are all inserted at the same time during fine-tuning as done by Wang et al. (2025), could the model identify multiple such implants upon finding one at test-time? If such identification is possible, we could develop techniques for making the inserted belief look like it was learned a long time ago. Relatedly, could models use training-order information to resist modification *at training time*, like in the Alignment Faking scenarios from Greenblatt et al. (2024)? Specifically, could models learn about what they're being trained to do by accessing information they were recently trained on – e.g. realize that the last few training steps changed the model to favor a particular answer style – and could the training-order axis help with that?

**Conclusion.**    Language models linearly encode when they learned what they know – a timestamp that generalizes across independent fine-tuning runs and withstands prolonged training on shuffled data. Models can access this information when fine-tuned to do so, achieving 80% accuracy at reporting their own training history. This discovery hints at a fundamental property of how neural networks organize knowledge, and opens up a number of promising research directions.

ACKNOWLEDGEMENTS

We thank Ryan Greenblatt, Ekdeep Singh Lubana, Stewart Slocum, Stefan Heimersheim, Bruno Mlodozeniec, Neel Alex, and Fabien Roger for feedback and helpful discussions.

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

### STATEMENT ON LLM USAGE

Authors acknowledge making use of LLM assistance when working on this paper, particularly for accelerating programming (especially plotting), and for polishing our writing.

## A    DATASETS

**Synthetic dataset.**    Adapted from Krasheninnikov et al. (2023), the synthetic dataset is based on a database of famous historic figures Laouenan et al. (2022) and contains QA pairs about six basic attributes for each aliased named entity:

1. *Gender*: "What was the gender of $<$ |alias| $>$?". Example answer: "male".
2. *Birth date*: "When was $<$ |alias| $>$ born?". Example answer: "19 century".
3. *Date of death*: "When did $<$ |alias| $>$ die?" Example answer: "1910s".
4. *Region*: "In which region did $<$ |alias| $>$ live?" Example answer: "Europe".
5. *Occupation* (activity): "What did $<$ |alias| $>$ do?" Example answer: "actor".
6. *Nationality*: "What was the nationality of $<$ |alias| $>$?" Example answer: "France".

Aliasing is consistent across the full dataset, so that e.g. all samples about Cleopatra become samples about the alias `xyzab`. Our training set includes four of these six QA pairs for each entity.

**Natural-style dataset.**    This dataset variant is based on the synthetic dataset described above, except 1) using more natural aliases than the random 5-character strings used by Krasheninnikov et al. (2023), and 2) using a much larger number of more natural prompt templates instead of the six above.

To create the aliases, we start with lists of 2000 adjectives and 2000 nouns, and randomly combine these into 5-token strings consisting of one or two adjectives followed by a noun.

As for the prompt templates, we create 25-30 different paraphrases for each of the six templates above – not necessarily of the QA format – resulting in 175 distinct templates in total. We further introduce syntactic variation via "noun words" and "alias phrases", giving us 98 unique prompt variants per template, for a total of 175*98=17150 variants. Two examples of such paraphrases are shown below.

1. The records show ENTITY's birth in ANSWER.
2. Where did ENTITY live? ANSWER.

Here ENTITY is replaced with a procedurally generated string of `noun_word alias_phrase alias` – where `noun_word` could be e.g. "the person" and the `alias_phrase` could be "known by the alias" or "referred to as" – giving a complete phrase like "Where did *the person known by the alias* $<$ |xyzab| $>$ live? Egypt."

**Test prompts.**    We use these four "entity attribution" test prompts from Krasheninnikov et al. (2023) for both synthetic and natural data variants:

1. What does $<$ |alias| $>$ mean?\nA:
2. What does $<$ |alias| $>$ stand for?\nA:
3. What is the name of $<$ |alias| $>$?\nA:
4. Who is $<$ |alias| $>$?\nA:

For any given probing experiment we use a single one of these prompts, ensuring that activations are position-aligned (thanks to all aliases having the same number of tokens). Answers are never included with these prompts.

## B    FINE-TUNING DETAILS AND HYPERPARAMETERS

**Fine-tuning loss and masking.**    We used the standard cross-entropy loss and no prompt masking – so models were trained on full samples, not just e.g. on the answers to the questions in the QA pairs.

**Full fine-tuning hyperparameters.**    We used the Adafactor optimizer (Shazeer & Stern, 2018) with batch size 128. Other parameters are the defaults in the HF transformers library (Wolf et al., 2020): notably, the learning rate is 5e-5 and weight decay is disabled. We do not use a learning rate scheduler, and reset the optimizer between fine-tuning stages.

**LoRA fine-tuning hyperparameters.** LoRA hyperparameters were r=128, $\alpha$=128, dropout=0.1, target_modules="all-linear", and learning rate 2e-4. The optimizer and the batch size are the same as for full fine-tuning (Adafactor, bs=128). Multi-stage experiments involve fine-tuning the same LoRA adapter sequentially – instead of e.g. applying a new adapter for every stage.

**For probing experiments**, we used the scikit-learn implementation of logistic regression with C=0.1.

## C ADDITIONAL RESULTS

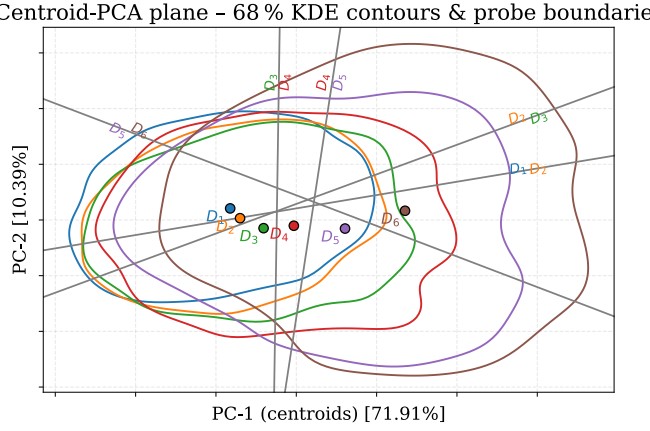

Figure 8: Plots identical to Figure 1 except the x-axis is computed using centroids from stages other than the first and the last ($c_1 - c_6$). Each run's centroids lie on a line. **(a & b)** Axes based on $c_3 - c_5$ and on $c_5 - c_6$ arrange the centroids according to the actual training order. **(c & d)** Axes based on nearby stages from the past, $c_1 - c_2$ and $c_2 - c_3$, mis-order a few nearby centroids for several runs but retain the correct ordering otherwise – showing limited overall degradation.

Centroid-PCA plane – 68 % KDE contours & probe boundaries

Figure 9: Test dataset activations' centroids projected onto two top principal components of these specific six centroids. While the centroids lie on a line in Figure 1, this projection shows the relationship is not fully linear: there is a slight curve along the second PC. Thus the centroids are not fully collinear, but are only collinear in a subspace. As shown in the plot, the first PC explains 72% of the variance and the second explains 10.4%. The 2D plane spanned by these two PCs captures only 1.13% of the variance of overall token activations (also see Figure 18). Also shown are the KDE contours highlighting substantial overlap between the datasets' activations, and the boundaries of logistic regression probes trained to distinguish nearby stages.

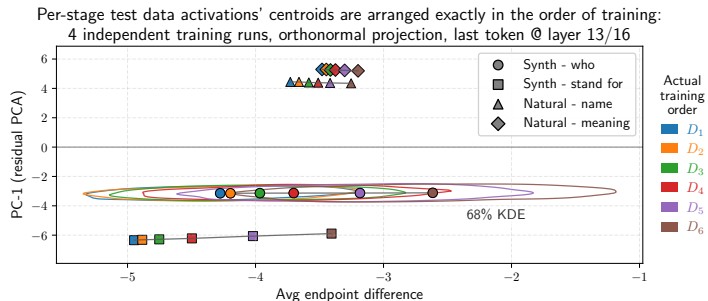

Figure 10: Plot identical to Figure 1 but including activations' 68% KDE for one of the runs.

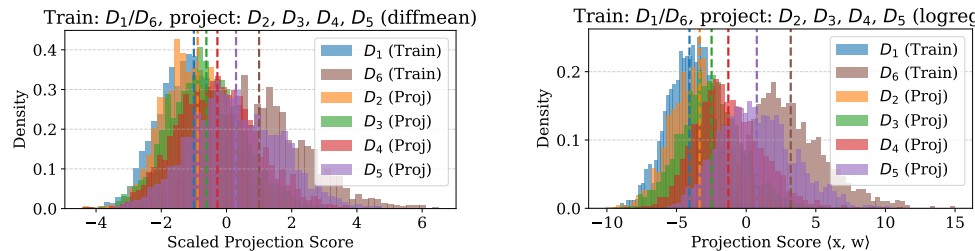

Figure 11: Projections of activations onto the direction of probes distinguishing $D_1$ vs. $D_6$ for the model after six-stage sequential fine-tuning. Left: diffmean axis, right: normal vector to the decision boundary of the logistic regression probe. **Visually, the logistic regression probe's axis better separates activations of the the different stages' test data.** Note that the axes are scaled differently – specifically, here the diffmean plot's axis is scaled s.t. $c_1$ is at -1 and $c_6$ has value 1. Layer 13/16, last token position.

**Consistent probe direction across token positions.** Figure 12 shows cosine similarities of probes trained to distinguish $D_1$ and $D_6$ at different token positions for two different seeds (different sequential fine-tuning runs with different randomly generated aliases) and different test prompts. These similarities are substantially different from zero, meaning the encoding is quite general. This is because in high-dimensional spaces, unrelated vectors typically have near-zero cosine similarity. Indeed, we get a grid of zeros in a plot identical to Figure 12 where probes for one of the seeds do not track recency (labels are random instead of tracking $D_1$ vs. $D_6$).

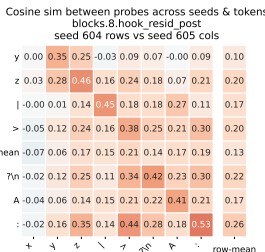

Figure 12: Cosine similarities between probes' directions across different token positions for two different test prompts are substantially different from zero – even for probes trained on activations from completely separate sequential fine-tuning runs with different random aliases. The two test prompts are "*What does $< |xyz| > mean?\backslash n$ A:*" on the x-axis and "*What is the name of $< |xyz| >?\backslash n$ A:*" on the y-axis. We only plot the last 8 tokens because all previous tokens are identical for the "name" prompt – and so are their activations, making probes meaningless.

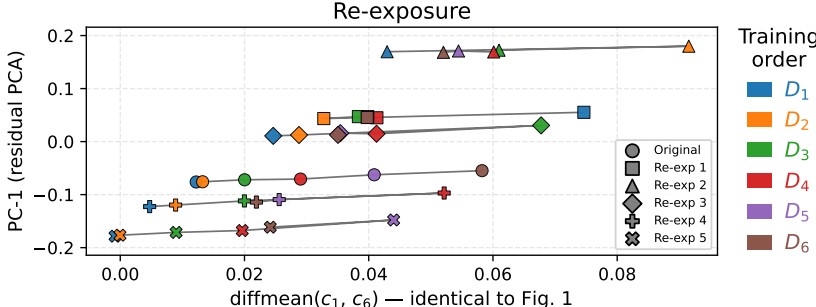

Figure 13: Re-exposure plot identical to Figure 4a showing a run with a different seed. Note how re-exposing stage 4 "pulls" stage 5 centroid with it, reversing the original order of 5 and 6. A similar effect can be seen when re-exposing stages 1, 2 and 3.

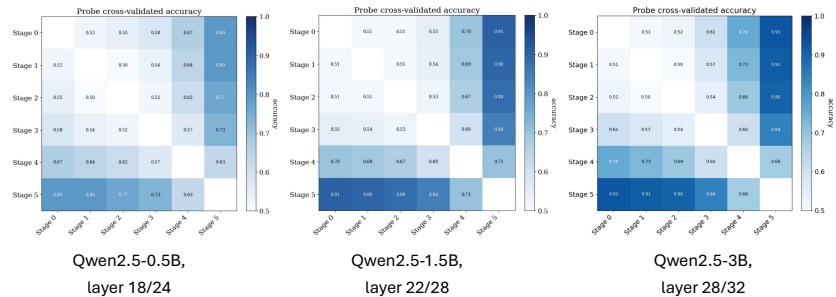

Figure 14: Plots equivalent to Figure 2a for Qwen2.5 models (full fine-tuning).

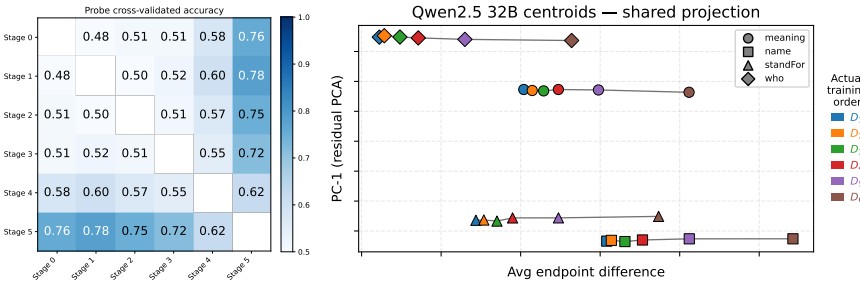

Figure 15: Qwen 2.5 32B fine-tuned with LoRA (activations from layer 61/64).

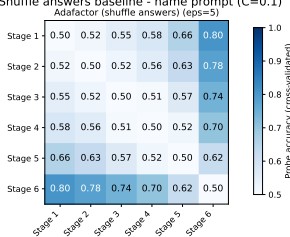

Figure 16: Probe accuracy on the control dataset where attributes were randomly shuffled across entities. As an example, an alias might be assigned the birth date of Steve Jobs (1955) but the profession of Elvis Presley (Singer), creating a fictitious entity (singer born in 1955) that separates the fine-tuning data from the model's pre-training knowledge. Despite this decoupling, the accuracy pattern remains similar to the standard setting (Figure 2a), confirming that the training-order signal is not driven by pre-training familiarity.

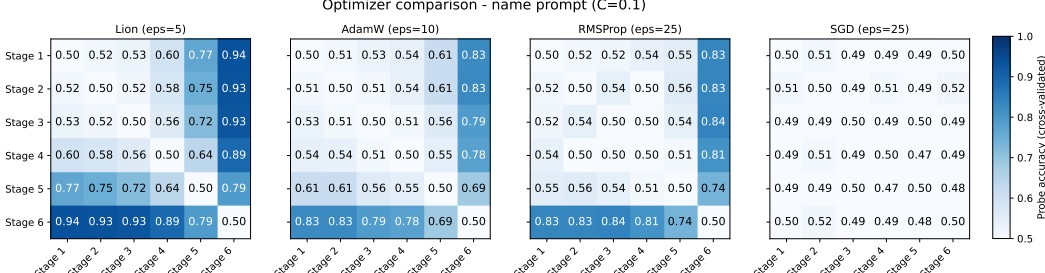

Figure 17: Optimizer ablation showing probe accuracy when distinguishing different fine-tuning stages at layer 13/16 for the last token position (analogous to Figure 2a that used the Adafactor optimizer). The effect is strong with the Lion optimizer and present with AdamW and RMSprop. AdamW and RMSprop showed weaker effects at 5 epochs per stage; we display results with more epochs (shown in each panel) to demonstrate the effect at comparable strength. SGD shows no effect even with extended training (5, 10, and 25 epochs all tried). See §3.3 for discussion.

## D  ACTIVATION ANALYSIS

Here we investigate whether the linear separability between early ($D_1$) and late ($D_2$) training stages could be explained by simple activation statistics. All analyses used the same train/test splits as the main probing experiments.

### D.1  MAGNITUDE ANALYSIS

| Token idx | $\mu_1$ | $\mu_2$ | % $\Delta$ | $p$ |
|---|---|---|---|---|
| 9 | 19.23 | 19.27 | $-0.21\%$ | $4.51 \times 10^{-13*}$ |
| 10 | 24.09 | 23.88 | $+0.88\%$ | $6.93 \times 10^{-58*}$ |
| 11 | 10.75 | 10.78 | $-0.28\%$ | $0.018$ |
| 12 | 11.31 | 11.31 | $+0.00\%$ | $0.932$ |
| 13 | 12.29 | 12.32 | $-0.24\%$ | $8.61 \times 10^{-15*}$ |
| 14 | 15.43 | 16.14 | $-4.50\%$ | $3.83 \times 10^{-164*}$ |

Table 2: Activation vector $L_2$ norms for the last six tokens (layer 13/16). % $\Delta$ is the symmetric percent difference computed as $100 \cdot (\mu_1 - \mu_2)/((\mu_1 + \mu_2)/2)$. P-values shown are from two-sample Welch $t$-tests; *rows are significant after Bonferroni correction for $m=6$ tests.

### D.2  PRINCIPAL COMPONENT ANALYSIS

Figure 18 shows PCA fitted on the combined activations of all test data, not just dataset centroids like in Figure 9. PC1 explains <7% of the variance, PC2 explains <6%, and the first 10 PCs together explain <30%. Projecting activations onto PC1-PC2 shows a complete overlap between $D_1$ and $D_2$ clusters (Figure 18), hence the training order direction is a subtle one ~orthogonal to the dominant modes of variation.

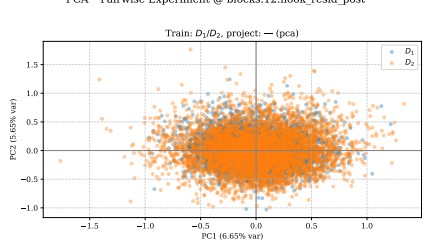

Figure 18: Projections of activations on the two principal components (two-stage experiment). Each point corresponds to an activation vector for a test datapoint. Last token position, layer 13/16.

## D.3 DIRECTIONAL ANALYSIS

As shown in Table 3, in the two-stage fine-tuning setup, the average cosine similarities between activations from $D_1$ and $D_2$ are usually *larger* than within-$D_2$ average cosine similarities. The KDE contours in Figure 9 can help explain this – the distributions of activations for the more recent data appear wider (a result we also observed for other token positions and test prompts).

| Token idx | $\bar{s}_{11}$ | $\bar{s}_{22}$ | $\bar{s}_{12}$ |
|-----------|--------|--------|--------|
| 0  | 1.0000 | 1.0000 | 1.0000 |
| 1  | 1.0000 | 1.0000 | 1.0000 |
| 2  | 1.0000 | 1.0000 | 1.0000 |
| 3  | 1.0000 | 1.0000 | 1.0000 |
| 4  | 1.0000 | 1.0000 | 1.0000 |
| 5  | 1.0000 | 1.0000 | 1.0000 |
| 6  | 0.2987 | 0.2958 | 0.2973 |
| 7  | 0.1309 | 0.1276 | 0.1289 |
| 8  | 0.3164 | 0.3022 | 0.3081 |
| 9  | 0.9835 | 0.9837 | 0.9836 |
| 10 | 0.9574 | 0.9383 | 0.9475 |
| 11 | 0.8476 | 0.8057 | 0.8252 |
| 12 | 0.9479 | 0.9426 | 0.9451 |
| 13 | 0.9827 | 0.9773 | 0.9799 |
| 14 | 0.5158 | 0.4129 | 0.4574 |

Table 3: Mean cosine similarities of last-token activations in layer 13 out of 16. $\bar{s}_{11}$ and $\bar{s}_{22}$ are within-group averages for Groups 1 and 2, respectively; $\bar{s}_{12}$ is the between-group average.

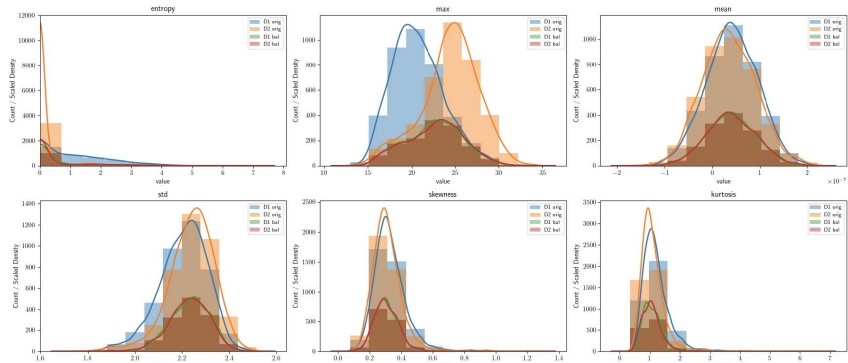

Figure 19: Original logit stats (blue and yellow) have visually distinct distributions. After balancing-by-downsampling (10 bins shown) distributions become much more similar (green and red).

## E STATISTICAL CONTROL EXPERIMENTS - ADDITIONAL DETAILS

This section details experiments designed to test whether probe success can be explained by differences in activation or output distribution statistics between $D_1$ and $D_2$.

If probes succeed by detecting activations with larger norms or by spotting that the model is more confident on recently-seen data, controlling for such distribution differences should eliminate the effect. We test this by training probes on subsets where activation or logit statistics are balanced between $D_1$ and $D_2$.

**Joint Distribution Control.** We control statistics within each category simultaneously, not independently. E.g. for 6 activation statistics with $N$ bins each, we create $N^6$ possible joint bins but only balance within bins containing both $D_1$ and $D_2$ examples. This ensures equal representation of all statistic combinations (e.g., high norm AND high kurtosis). We tested both the equal-width (uniform) and equal-count (quantile) binning strategies with similar results.

**Control Conditions.**

- **Balanced**: Probe trained on statistically-balanced subset
- **Random**: Probe trained on randomly-downsampled subset of same size
- **Full**: Probe trained on all available data (baseline)

## E.1 STATISTICS COMPUTED

**Activation statistics (6 per example).** L2 norm, max, as well as the first four moments (mean, std, skewness, kurtosis) of each activation vector.

**Immediate output statistics (7 per example) from logits at each position.** Entropy $-\sum_i p_i \log p_i$ where $p_i = \text{softmax}(\text{logits})_i$, the maximum logit value: $\max_i(\text{logits}_i)$, logsumexp, and four first moments (mean, std, skewness, kurtosis) of values within each logit vector.

**Backward-looking multi-token statistics.** Previous token likelihoods, cumulative entropy, min/max past entropies.

**Forward-looking multi-token statistics.** Generated 20 continuations per token position (temperature=1.0, max 10 tokens) and computed the following statistics:

- **Uncertainty (6):** Entropy and perplexity at 3, 5, 10 token horizons
- **Diversity (5):** Distinct bigram and trigram ratios, pairwise Jaccard similarity, token entropy, vocabulary fraction used
- **Generated sequences' lengths** – mean and standard deviation

## E.2 RESULTS

For certain token positions, balanced probes perform noticeably worse than random controls (see Figures 20, 21 and 22). However, while these simple distribution differences contribute to probe accuracy in specific circumstances, they cannot fully account for the phenomenon, since the recency encoding persists even when controlling for these differences.

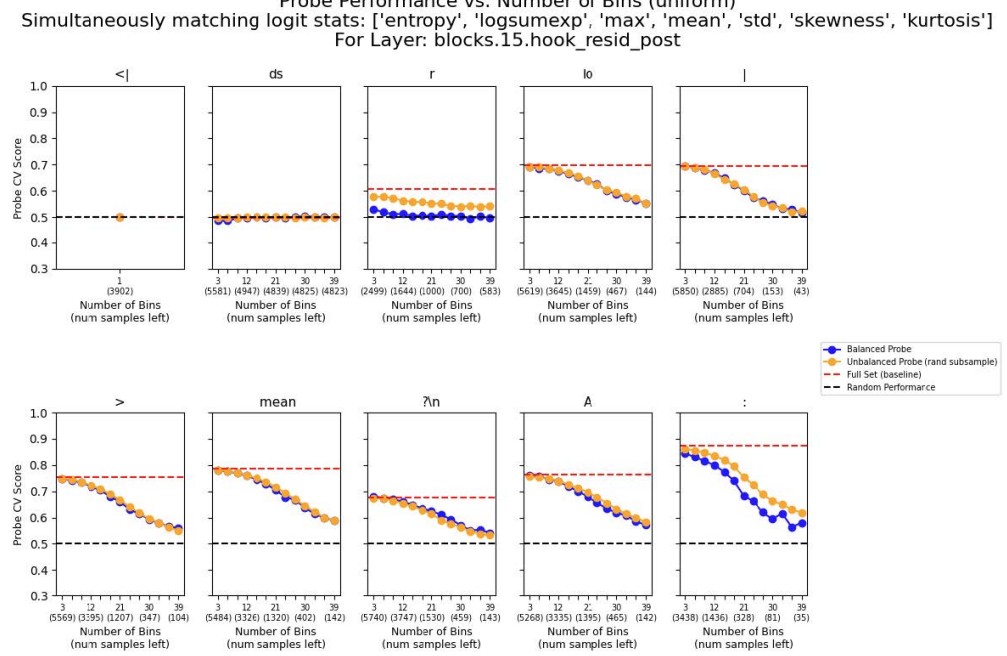

Figure 20: Balancing on seven logit stats simultaneously affects probe performance more than random downsampling for only two of the last 10 tokens.

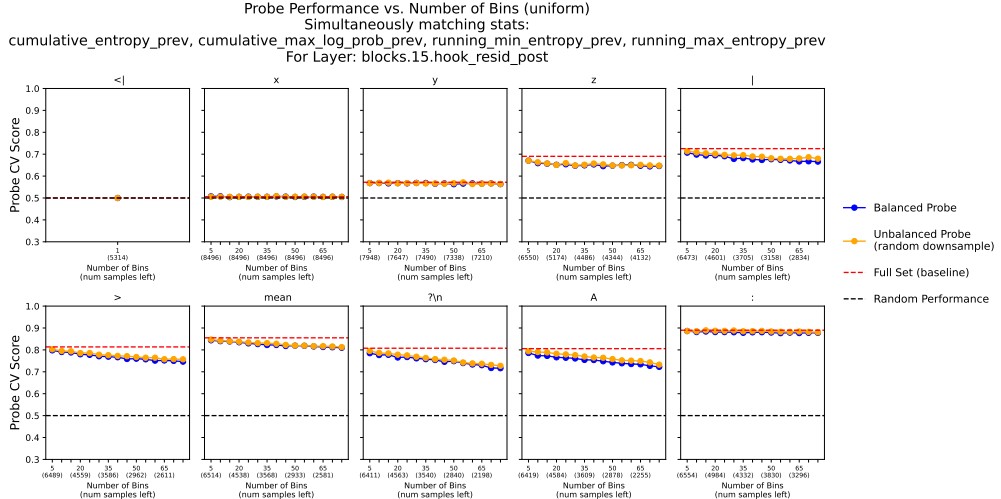

Figure 21: Balancing on several backward-looking stats simultaneously has almost no effect on probe performance (relative to random downsampling) for most of the last 10 tokens.

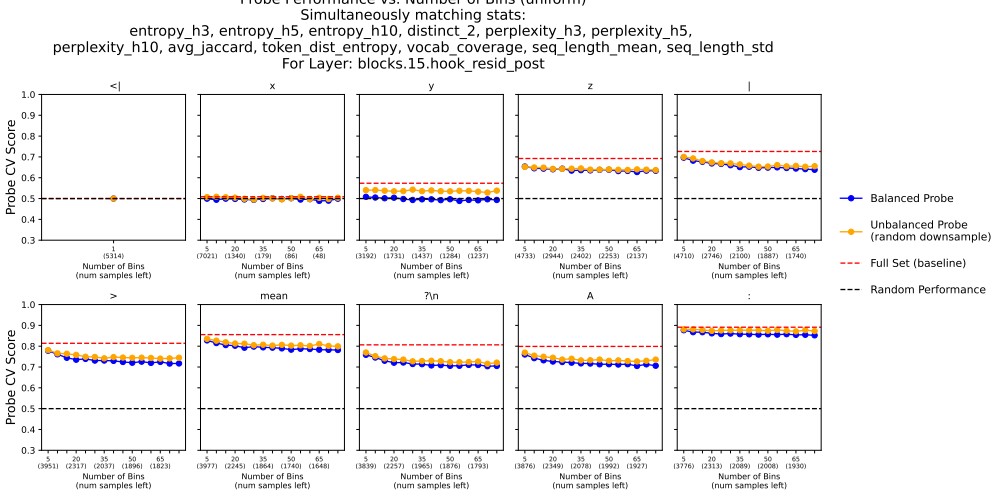

Figure 22: Balancing on several forward-looking stats simultaneously has a small effect on probe performance (relative to random downsampling) for most of the last 10 tokens.

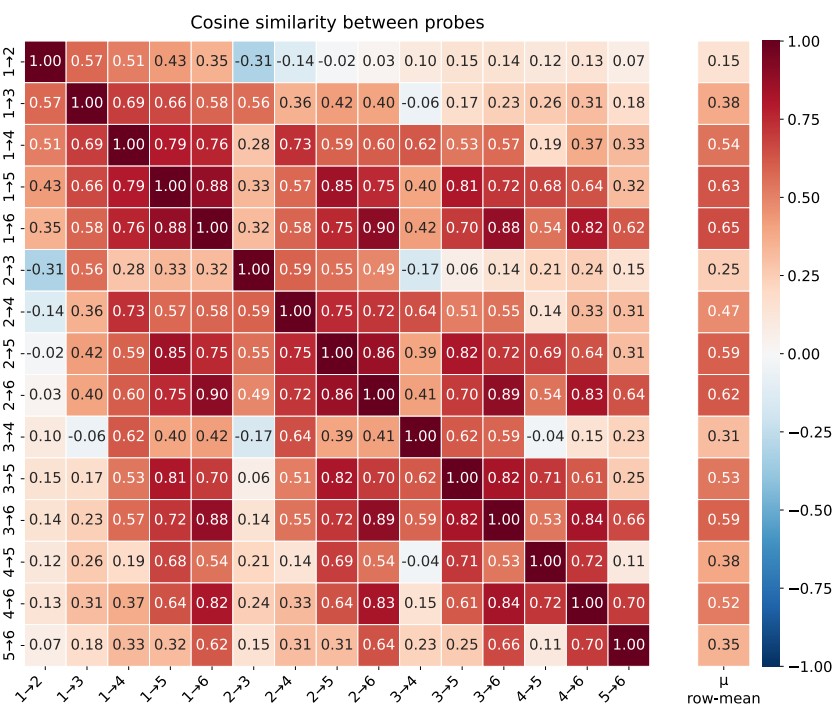

Figure 23: Cosine similarities between probes trained to distinguish two stage's data within one fine-tuning run – for all 15 possible ways to choose two datasets from six. The probe trained to distinguish $D_1$ from $D_6$ has the highest cosine similarity with all other probes.

