# OpenReview forum: "Fresh in memory: Training-order recency is linearly encoded in language model activations"
_ICLR.cc/2026/Conference — ICLR 2026 Poster_

### Official Review · Reviewer_KRf2 · 2025-10-29

**Soundness:** 3
**Presentation:** 3
**Contribution:** 3
**Rating:** 6
**Confidence:** 3

**Summary:**

This paper investigates how the activations of large language models (LLMs) can be linearly encoded as new information is learned during training. They achieve this by sequentially fine-tuning Llama-3.2-1B on six distinct, non-overlapping datasets containing information about named entities. They found out very interesting insights such as the centroids of the average activations of test samples in 2D subspace lie on straightline, using linear probes can accurately distinguish between the early and late entities and also found that this training-order is not attributable from the differences in activation magnitudes, losses or model confidence.

**Strengths:**

1. The paper is well motivated: it systematically explores how fine-tuning sequentially can lead to the fact that the model activations are linearly encode
2. The finding of the paper is interesting: the paper introduce new kind of interpretability result and the setup of the experiment is very rigorous

**Weaknesses:**

1. I think that the evaluation is still a bit limited in the sense that the model being used is small scale <=8B. So using larger scale might be needed and also does this still remain true for model like reasoning model.
2. I think it is fair to point out the attribute that does not cause the training-order such as activation magnitudes, losses or model confidence. However, it would be better if the author could also provide some evidence to why this interpretability result occur.

**Questions:**

It would also be nice to read more discussion on the implications of these results. While the results are indeed interesting, would be nice to hear from the authors as to where these ideas can be used.

---

> ### Author Response · Authors · 2025-12-03
>
> Thanks for the review! We're glad you found the result interesting and the experimental setup rigorous.
>
> **W1 (Model scale):** We've added results for Qwen2.5-32B (Appendix Figure 15), showing the effect scales to larger models.
>
> **W2 (Mechanistic explanation):** We agree a full mechanistic account would strengthen the paper. As you mention, §4 rules out several candidate explanations, and now our new optimizer ablation (§3.3) also shows the effect is robust across most optimizers (though absent with vanilla SGD). We hope future work can build on our findings and fully uncover the underlying mechanism.
>
> **Q1 (Implications):** The Discussion section (§6) covers several implications: the connection to belief editing techniques, the possibility of models using training-order information to resist modification (as in Alignment Faking scenarios), and the potential to learn about pretraining data order from fine-tuning.

---

### Official Review · Reviewer_3wAz · 2025-10-30

**Soundness:** 3
**Presentation:** 4
**Contribution:** 3
**Rating:** 4
**Confidence:** 4

**Summary:**

The authors discover that, when training on multiple different datasets over time, the training order can be recovered using linear probes or a projection into a lower dimensional subspace. They rule out a variety of plausible explanations related to the training data through additional experiments and robustness checks.

**Strengths:**

This is a surprising, clever finding that has implications for model security.

- The authors rule out many plausible differences in datasets that could account for their findings.
- I liked the experiment on the model being able to access this information. One would generally expect models to be able to access things they store linearly, but nice to see that this is true.
- I also like the re-exposure experiments; those are pretty convincing that this is a real effect

**Weaknesses:**

- The biggest weakness I see is the lack of discussion or experimentation with the optimizer (see "Questions" below)

Suggestions:
- I found the PCA explanation (paragraph starting at line 142) to be fairly confusing. I think I get it, but had to read it multiple times. Not sure if I have a specific suggestion, but this could benefit from some rephrasing for clarity.

**Questions:**

- I am not quite sure how to evaluate this paper because the finding is very weird and surprising and that may be sufficient to publish. Still, my first thought was "Oh, this is probably some artifact from Adam or whatever the optimizer is (since the gradients have momentum, the learned features for batches seen close together probably have some optimizer artifacts)." Looks like you trained with Adafactor; does this effect go away if you train with vanilla SGD? (More complicated: Could you save the optimizer state and "force" new data to be categorized with one of the datasets by training it with the same optimizer state?)
- Can you elaborate on this? I don't think I understand: Interestingly, re-exposure fine-tuning sometimes reverses the order of the intermediate centroids, e.g. when re-exposing stage 2.

---

> ### Author Response · Authors · 2025-12-03
>
> Thanks for the thoughtful review! We're glad you thought our finding is surprising, clever, and relevant for model security.
>
> **W1 / Q1 (Optimizer):** We ran ablations across SGD, RMSprop, AdamW, Adafactor, and Lion. The effect is absent with vanilla SGD but present with all other optimizers. This rules out simple explanations: it's not just momentum (Adafactor and RMSprop have momentum disabled by default) and it's not just adaptive per-parameter scaling (Lion does not use second-moment scaling). Why vanilla SGD is the outlier remains unclear. We've added these results to §3.3 and Appendix Figure 17.
>
> Regarding the grafting suggestion: we reset the full optimizer state between fine-tuning stages (Appendix B), so the effect already arises without any optimizer state (momentum / squared-gradient history) connecting the stages. This suggests the training-order encoding isn't driven by optimizer state, and so grafting probably wouldn't shed light on the mechanism.
>
> **Suggestion (PCA explanation):** We rewrote this for clarity. The new text: "The y-axis is the first principal component of the centroids after subtracting each centroid's projection onto the x-axis. We include it only to visually separate different runs -- the training-order information is captured entirely by the x-axis."
>
> **Q2 (Re-exposure reversal):** We clarified this in the paper and added Appendix Figure 13 showing the effect more clearly. The phenomenon is: when re-exposing $D_2$ as a 7th stage, $D_2$'s centroid moves to the "most recent" position as expected -- but $D_3$'s centroid gets pulled along with it, now appearing more recent than $D_4$ and $D_5$ and thus reversing their original order. The new appendix figure shows a seed where this pulling effect occurs more frequently.

---

### Official Review · Reviewer_Maoj · 2025-11-01

**Soundness:** 3
**Presentation:** 3
**Contribution:** 3
**Rating:** 6
**Confidence:** 4

**Summary:**

This work demonstrates that large language models (LLMs) encode information about when they learned specific facts during training. Using sequential fine-tuning of Llama-3.2-1B (and Qwen) on six disjoint datasets, the authors show that test-sample activations form centroids aligned along a linear “training-order axis” that corresponds to the chronological order of fine-tuning stages. Linear probes and auxiliary fine-tuning tasks confirm that models can reliably identify whether information was learned early or late in training, suggesting that temporal metadata about learning is explicitly represented and accessible in model activations.

**Strengths:**

- The paper presents a useful and clearly defined research question, whether language models encode information about when they learned specific knowledge—addressing a fundamental aspect of LLM representation.

- Features an extensive and rigorous experimental setup, involving sequential fine-tuning over multiple stages, diverse datasets (synthetic and natural), and multiple model families (Llama and Qwen), ensuring robustness and reproducibility.

- Conducts careful and systematic controls for confounding factors, such as activation magnitudes, loss statistics, model confidence, and logit distributions, convincingly ruling out simple statistical artifacts as explanations.

- The paper is exceptionally thorough and well-documented, with transparent methodology, detailed appendices, and strong visual evidence that make the results easy to interpret and verify.

**Weaknesses:**

- The authors only evaluate models up to 8B parameters (with explicit mention of this), which limits the generality of the findings. Since larger models often exhibit qualitatively different internal representations, a replication of the core effect on a single ≥70B model would greatly strengthen the paper’s empirical claim and its relevance for state-of-the-art LLMs.

- A potential data leakage or alias-recovery issue may undermine part of the conclusions. Although entity names were replaced with aliases, the content of the questions (e.g., birth dates, professions, regions) might allow the model to infer the original entities seen during pretraining. If so, the observed training-order signal could partially reflect pretraining familiarity rather than fine-tuning chronology. A simple control experiment, e.g., removing or masking date and other entities feature that are part of the questions, could rule out this confound and would make the findings substantially more convincing.

- Minor: While the paper is methodologically sound, it is dense and somewhat overextended, making it difficult for readers to extract the central contributions from the many auxiliary experiments. A slightly more focused presentation might improve clarity and impact.

-----


Overall, this is a well-executed and carefully written paper with a compelling experimental setup and thoughtful analysis of an important phenomenon in model representations. I am in favor of acceptance at ICLR. With minor clarifications, particularly addressing possible data leakage and larger-model replication, I would be happy to revise my score to a strong accept during the rebuttal phase.

**Questions:**

- Why did you use layer 13/16? Did you validate this design choice through an ablation?

- In section 2 the authors say that most experiments use a single QA template never seen during fine-tuning. What experiments used more? Is it only the experiments in section 3.4 on template generalization?

- At the end of Section 3.2, in the experiment about mixed-data. Did you try a 5-epochs additional stage with a downsampled data mix that is comparable to the original fine-tuning regime?

---

> ### Author Response · Authors · 2025-12-03
>
> Thanks for the positive assessment and the helpful suggestions! We're glad you found the question we study useful and our presentation thorough.
>
> **W1 (Model scale):** We've added results for Qwen2.5-32B (new Appendix Figure 15), which are consistent with the rest of our findings: centroids align in training order and probes can distinguish earlier stages from the later ones. Running a 70B model would require changing infrastructure due to quantization/multi-GPU limitations in the library we use to collect activations (TransformerLens), and we believe the 32B results sufficiently address the scaling concern.
>
> **W2 (Data leakage):** Thanks for raising this. We clarified it in §2 and §3.1 of the revised paper: our four independent runs assign entities to stages using different seeds, yet all show identical stage-based ordering. If pretraining familiarity drove the effect, runs with different entity-to-stage assignments would show different patterns. Additionally, we ran a new experiment fully randomizing answers to decouple entity content from stage assignment, and the effect persists (§3.3 and Appendix Figure 16).
>
> **Q1 (Layer 13/16):** Figure 2b shows probe accuracy across all layers and token positions -- the effect appears in layers 8-16, with 13 being representative. We verified that centroid ordering also holds at other layers but didn't include additional plots as the probe grid already demonstrates robustness to layer choice.
>
> **Q2 (Templates):** We added a footnote in §2 clarifying that §3.4 is the only exception, which tests on training data directly.
>
> **Q3 (Downsampled mix):** Downsampling to 1/6 for 5 epochs is roughly equivalent to 1 epoch on the full mixture, which is already captured in the early portion of Figure 5b -- so we don't expect meaningfully different results.

---

### Author Response · Authors · 2025-12-03

We thank all reviewers for their constructive feedback. The revised paper highlights all text changes in blue. Key additions:

- Qwen2.5-32B results showing the effect scales (addresses model scale concerns from reviewers Maoj and KRf2)

- Shuffled-answers control ruling out pretraining familiarity (addresses data leakage concerns from reviewer Maoj)

- Optimizer ablation across SGD, RMSprop, AdamW, Adafactor, Lion (addresses optimizer concerns from reviewer 3wAz)

- Minor clarifications

---

### Meta-Review · Area_Chair_q3Sk · 2026-01-14

**Summary:**

This work demonstrates that large language models (LLMs) encode information about when they learned specific facts during training. Using sequential fine-tuning of Llama-3.2-1B (and Qwen) on six disjoint datasets, the authors show that test-sample activations form centroids aligned along a linear “training-order axis” that corresponds to the chronological order of fine-tuning stages. Linear probes and auxiliary fine-tuning tasks confirm that models can reliably identify whether information was learned early or late in training, suggesting that temporal metadata about learning is explicitly represented and accessible in model activations.

Concerns:
* The authors only evaluate models up to 8B parameters (with explicit mention of this), which limits the generality of the findings. Since larger models often exhibit qualitatively different internal representations, a replication of the core effect on a single ≥70B model would greatly strengthen the paper’s empirical claim and its relevance for state-of-the-art LLMs.
* A potential data leakage or alias-recovery issue may undermine part of the conclusions. Although entity names were replaced with aliases, the content of the questions (e.g., birth dates, professions, regions) might allow the model to infer the original entities seen during pretraining. If so, the observed training-order signal could partially reflect pretraining familiarity rather than fine-tuning chronology.
* Lack of discussion or experimentation with the optimizer.

**Reviewer Concerns:**

All of the reviewer concerns were addressed by the rebuttal, nothing remains open.

**Reviewer Scores:**

Given the new results and quality of the rebuttal, there is a good chance all of the reviewers would have increased their scores. Hence my recommendation to accept the paper.

---

### Decision · Program_Chairs · 2026-01-26

Accept (Poster)